# Enabling Synergistic Full-Body Control in Prompt-Based Co-Speech Motion Generation

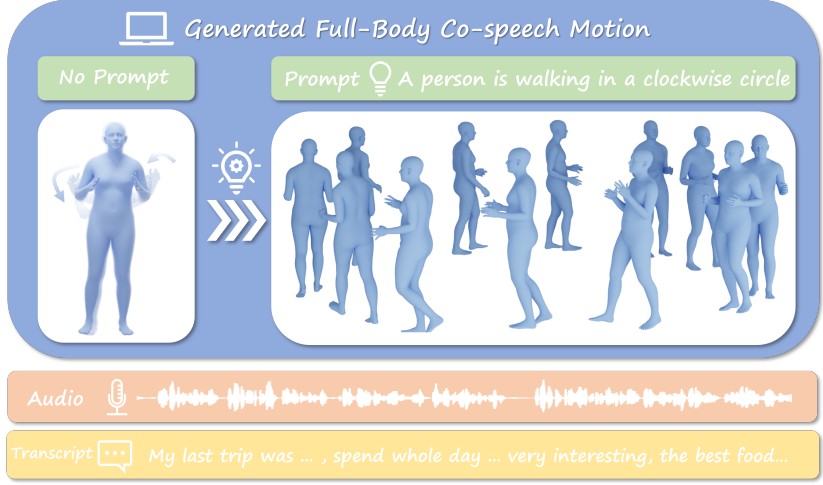

**Figure 1: Given audio and script of speech, as well as *arbitrary motion-related text prompt*, our method can generate full-body synergistic motion matching both speech content and prompt even if the motion is unseen in the speech-to-motion dataset used for training, such as the "walking in a clockwise circle" example in the figure. Meanwhile, the generation result is also highly consistent with the script content and the audio rhythm of the input speech.**

## ABSTRACT

Current co-speech motion generation approaches usually focus on upper body gestures following speech contents only, while lacking supporting the elaborate control of synergistic full-body motion based on text prompts, such as *talking while walking*. The major challenges lie in 1) the existing speech-to-motion datasets only involve highly limited full-body motions, making a wide range of common human activities out of training distribution; 2) these datasets also lack annotated user prompts. To address these challenges, we propose *SynTalker*, which utilizes the off-the-shelf text-to-motion dataset as an auxiliary for supplementing the missing full-body motion and prompts. The core technical contributions are two-fold. One is the multi-stage training process which obtains an aligned embedding space of motion, speech, and prompts despite the significant distributional mismatch in motion between speech-to-motion and text-to-motion datasets. Another is the diffusion-based conditional inference process, which utilizes the separate-then-combine strategy to realize fine-grained control of local body parts. Extensive experiments are conducted to verify that our approach supports

precise and flexible control of synergistic full-body motion generation based on both speeches and user prompts, which is beyond the ability of existing approaches. The code is released on (link will be published upon acceptance).

## CCS CONCEPTS

• **Computing methodologies** → **Motion processing**; **Computer graphics**.

## KEYWORDS

co-speech motion generation, text-to-motion generation, vector quantization, diffusion model.

## 1 INTRODUCTION

Co-speech motion generation [9, 10, 12, 27, 48, 56], which generates stylized movements of human body following speech audio inputs, is among the central tasks in creating digital talking avatars. Though growing rapidly in recent years, current co-speech motion generation approaches usually focus on upper-body gestures, such as head and hands, or only support limited full-body motions, in special restricted low-body movements. One of the fundamental challenges here is that the speech signal is too weak to uniquely determine full-body motions. For example, for generating co-speech motion of a digital host for releasing a new product, both "talking while walking" and "talking while standing still" are reasonable motions. As a result, it would be meaningful to realize precise and flexible control of full-body motion for achieving natural and

*ACM MM, 2024, Melbourne, Australia*

© 2024 Copyright held by the owner/author(s). Publication rights licensed to ACM.
ACM ISBN 978-x-xxxx-xxxx-x/YY/MM
https://doi.org/10.1145/nnnnnnn.nnnnnnn

synergistic effects based on additional input signals to reflect user intentions, such as text prompts.

On the other hand, prompt-based co-speech motion generation is a highly nontrivial task with two major reasons. On one hand, the existing speech-to-motion datasets, such as BEATX [26], focuses on subtle hand movements yet involve fairly limited full-body motions, especially in lower body. For example, the lower body of the speaker usually remains relatively stable during talking. This makes a wide range of common human activities out of training distribution. On the other hand, these datasets also lack annotated user prompts. Furthermore, crafting of a diverse and annotated dataset at scale is extremely costly. This has significantly constrained the potential for quality and diversity in motion generation.

One possible solution to deal with the data lacking issue is to augment training with text-to-motion datasets, such as AMASS [30], which include a relatively complete set of full-body motions with vast scale and strong diversity, as well as annotated text prompts [16, 35]. Superficially, jointly training with both speech-to-motion and text-to-motion datasets could lead to the ideal model, whose key is to build a joint embedding space of speech, text, and motion. However, due to the significant distribution mismatch in motion between the two kinds of datasets, a large number of full-body motions are missing their corresponding speech signals, making building such an embedding space still a challenging task.

To deal with issue, we propose SynTalker, a prompt-based co-speech motion generation approach which utilizes off-the-shelf text-to-motion datasets to augment co-speech training, meanwhile addressing the distributional mismatch challenge. For training, we propose a multi-stage approach, which utilizes motion representation pre-training and motion-prompt alignment pre-training to address the issue of motion distribution mismatch and the problem of lacking prompt annotation for speech-to-motion data. For inference, we designed a novel separate-then-combine strategy under for both input conditions and body parts, such that the separate operations map the input signal to their most proper body part to control, meanwhile the combine operations leads to the synergy among body parts. Extensive experiments show that, our approach is able to achieve significant performance in using both speech and text prompt to guide the generation of synergistic full-body motion precisely and flexibly, which is beyond the capability of the existing co-speech generation approaches.

In summary, by proposing SynTalker, our main contributions are: 1) We propose the first approach to enable synergistic full-body control with general text prompts for co-speech motion generation, under the situation of lacking fully annotated datasets of speech, text, and motion; 2) We propose a novel multi-stage training approach to address the motion distributional mismatch and prompt annotation lacking challenges; 3) We propose a novel separate-then-combine approach for model inference to achieve both precise control and synergistic motion generation.

The rest of the paper is organized as follows. In Section 2, we discuss the related work from two closest research areas, i.e. co-speech motion generation and text-to-motion generation. In Section 3-5, we introduce the model design, training process, and inference process of our approach in detail. In Section 6, we report experimental results. In section 7-8, we discuss limitations and future work as well concluding the paper.

## 2 RELATED WORK

### 2.1 Co-Speech Motion Generation

Early rule-based approaches to co-speech motion generation [5, 6, 21] utilize linguistic rules to translate speech into sequences of predefined gesture segments. This process, being time-consuming and labor-intensive, requires significant manual effort in defining rules and segmenting motions. Previous generative methods often produce overly smooth motions [3, 12, 27, 40], attributable to the use of traditional deterministic generative models, which are inadequate for many-to-many mapping problems. Despite some attempts to introduce control signals and prior information into model design [1, 22, 27, 49], the capabilities of these models remain limited. Recent advancements have leveraged modern generative models like Diffusion [11] to tackle these challenges. For instance, DiffGesture [56] employs a diffusion model to capture the relationship between speech and gesture. Nonetheless, the weak semantic signals in audio often result in motions that are misaligned with the semantic content of the input audio. DiffuseStyleGesture [48] advances this by integrating emotional control into the gesture generation process, while Amuse [9] and EMOTE [10] explicitly extract and disentangle emotions from given conditions to provide stronger control signals. UnifiedGesture [47] additionally use reinforce learning to strength gesture. GestureDiffuCLIP [2] incorporates existing contrastive learning frameworks [41] to enable prompt-based gesture style control, offering finer-grained style controllability for end-users. However, these methods still struggle to meet diverse real-world user requirements, such as accommodating gestures while walking, due to the limited motion distribution in co-speech datasets.

### 2.2 Text-to-Motion Generation

In parallel to co-speech motion generation problems, text-based motion generation aims to generate general motions from textual prompts. Pioneering works [7, 29, 42, 54, 55] such as Motion Diffuse and T2M-GPT utilize a diffusion-based architecture or GPT-based architechture to model the many-to-many challenges in text-to-motion generation. Subsequent studies, such as PriorMDM, TLControl, and OmniControl [38, 45, 46], further employ trajectory and end-effector tracking to provide finer-grained control. GMD [20] introduces additional scene information during the generation of human actions, and MotionClip [41] attempts to align motions with the CLIP space [36], enabling the capability to generate motion from images. TM2D [14] and FreeTalker [50] have explored this by learning both speech-to-motion and text-to-motion tasks simultaneously. Even though this enables a single model to switch between two tasks, it does not provide synergistic generation conditioned on both signals.

## 3 MODEL DESIGN

In this section, we introduce our prompt-based co-speech motion generation model. We first provide an overview of the model design and the corresponding generation process. Afterwards, we provide detailed descriptions of two core modules for motion representation and conditional generation.

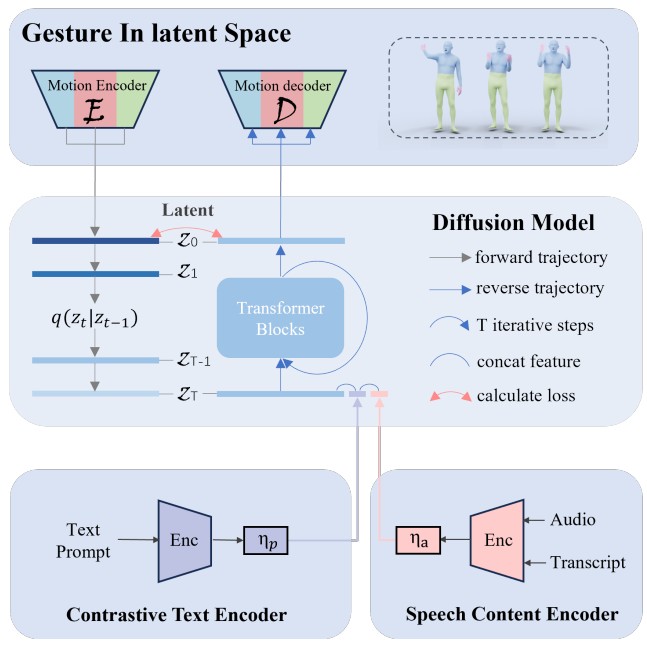

**Figure 2: The structure of our prompt-based co-speech generation model.**

## 3.1 Overview

Our model takes speech audio and the corresponding transcripts as inputs, targeting at outputting realistic and stylized full-body motions that align with the speech content rhythmically and semantically. Compared with traditional co-speech generation model, besides speech, it further allows to use a short piece of text, namely a *text prompt* to provide additional descriptions for the desired motion style. The full-body motions are then generated to follow the style given by both speech and prompt as much as possible.

The overall model structure is illustrated in Figure 2, which consists of three major components. The first is the *motion representation module*, which consists of motion encoder and decoders. We include three separate encoder and decoders to represent local body parts. The second is the *conditional generation module* for aligning latent motion representations with conditional inputs of speeches and prompts. The module is based on the latent diffusion model [37], which apply diffusion and denoising steps in the latent space. The third is the *conditional representation module*, which consists of the speech content encoder and contrastive text encoder to obtain scalar-valued prompt and speech conditions in the diffusion-based conditional generation model. Below we dive into detailed structures of the first two modules.

## 3.2 Motion Representation Module

**Motion encoding**. Recent studies in motion generation have demonstrated that vector-quantized autoencoder (VQ-VAE) [43] possesses a remarkable capability for compressing motion information [2, 15, 54]. We also utilize vector quantization for motion encoding. Following [15, 31, 53], we use a residual VQ-VAE (RVQ-VAE) as the quantization layer. To further decrease the coupling between body parts, we segment the body into three parts: upper body, fingers, and lower body, like in [2, 26], and train a separate RVQ-VAE for each part. In details, the motion sequence $\mathcal{M}$ can be represented as $\mathbf{m}_{1:N} \in \mathbb{R}^{N \times D}$, which is firstly encoded into a latent vector sequence $\mathbf{z}_{1:n} \in \mathbb{R}^{n \times d}$ with downsampling ratio of $n/N$ and latent dimension $d$, using 1D convolutional encoder E; The $\mathbf{z}_{1:n} \in \mathbb{R}^{n \times d}$ obtained through the encoder then enters the first quantization layer $Q_1$, each vector subsequently finds its nearest code entry in the layer's codebook $\mathbf{C_1} = \{\mathbf{c_k^1}\}_{k=1}^{K} \subset \mathbb{R}^d$ to get the first quantization code $\hat{\mathbf{z}}_{1:n}^1$, also we can calculate the quantization **residual**$_{1:n} = \hat{\mathbf{z}}_{1:n}^1 - \mathbf{z}_{1:n}$. The **residual**$_{1:n}$ then enter the second quantization layer $Q_2$ finds its nearest code entry in the layer's codebook $\mathbf{C_2} = \{\mathbf{c_k^2}\}_{k=1}^{K} \subset \mathbb{R}^d$ to get the second quantization code $\hat{\mathbf{z}}_{1:n}^2$. Accordingly, $\hat{\mathbf{z}}_{1:n}^3, \hat{\mathbf{z}}_{1:n}^4 \dots$ can be calculated in this manner. As the last step of motion encoding, we sum all quantization code together to get the final code $\hat{z} = \sum_{q=1}^{Q} \hat{z}^v$.

**Motion decoding**. Similar to motion encoding, three separated decoders are introduced for generating corresponding motions for all body parts, which are 1D convolutional decoders. During training, the motion data is encoded with motion encoders and fused with speech and prompt conditions by the diffusion-based conditional generation module, and then passed through the decoders to get the reconstructed motions. During inference, the motion encoder is not utilized, the generated motion is obtained directly from the speech and prompt conditions with the diffusion module and motion decoders.

## 3.3 Conditional Generation Module

The conditional generation module is based on the latent diffusion model [37], which is a variant of diffusion models that applies the forward and reverse diffusion processes in a pre-trained latent feature space. The *diffusion process* is modeled as a Markov noising process. Starting from a latent gesture sequence $Z_0$ drawn from the gesture dataset, the diffusion process progressively adds Gaussian noise to the real data until its distribution approximates $\mathcal{N}(0, I)$. The distribution of the latent sequences thus evolves as

$$q(Z_n|Z_{n-1}) = \mathcal{N}(\sqrt{\alpha_n}Z_{n-1}, (1 - \alpha_n)I), \qquad (1)$$

where $Z_n$ is the latent sequence sampled at diffusion step $n$, $n \in \{1, \dots, N\}$, and $\alpha_n$ is determined by the variance schedules. In contrast, the *reverse diffusion process*, or the *denoising process*, estimates the added noise in a noisy latent sequence. Starting from a sequence of random latent codes $Z_N \sim \mathcal{N}(0, I)$, the denoising process progressively removes the noise and recovers the original latent code $Z_0$. To achieve conditional motion generation, we train a network $E_\theta(Z_n, n, A, P)$, the *denoising network*, to recover the noise-free codes based on the noisy latent motion codes $Z_n$, the diffusion step $n$, the audio $A$, and the prompt feature $P$ from the joint align space. Finally, the recovered code is input into the motion decoders for motion generation.

## 4 MODEL TRAINING

The overall training pipeline is shown in Figure 3, which consists of a pre-training stage and a generation model training stage. The

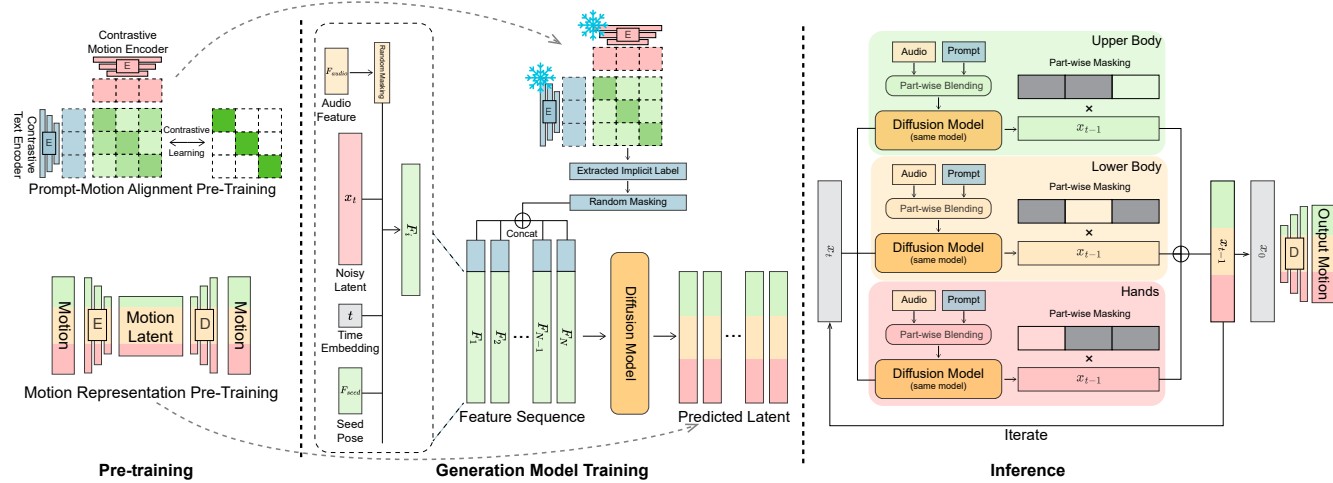

**Figure 3: Illustration of the training and inference processes. We initially train a contrastive learning space between text and motion, alongside a motion auto-encoder that uses motion from both speech-to-motion and prompt-to-motion dataset for an expressive latent space. Subsequently, our co-speech latent diffusion model is trained under the guidance of an *implicit label* extracted from motion using the contrastive space, effectively bypassing the lack of textual motion annotations in co-speech data. During inference, we implement a separate-then-combine strategy in every diffusion step, enabling finer control over individual body parts while preserving their synergistic interaction.**

pre-training stage involves two tasks. The first task, motion representation pre-training, targets at training the motion encoders and decoders for all body parts based on motion data from both the speech-to-motion and prompt-to-motion datasets in order to address the issue of motion distribution mismatch. The second task, prompt-motion alignment pre-training, targets at obtaining prompt-motion aligned embedding space [34, 41] based on the prompt-to-motion dataset and address the issue of lacking prompt annotations for speech-to-motion data. In the generation model training stage, both the speech-to-motion and prompt-to-motion data are utilized jointly with the motion encoders and decoders as well as the prompt-motion alignment space to obtain the final diffusion-based generation model [42]. Below we discuss the details of the training process.

### 4.1 Motion Representation Pre-Training

The target of motion representation pre-training is to obtain the motion encoders and decoders based on motion data only, which are from the speech-to-motion and prompt-to-motion datasets. By this way, the obtained primitive motion representation space are independent of any conditional speech and prompt signals. From extensive experiments, we find that this approach effectively alleviates the motion distribution mismatch issue between speech-to-motion and prompt-to-motion datasets, which can not be addressed when directly mixing the two datasets for conditional generation model training without such a pre-training process.

Concretely, utilizing all motion data from the two kinds of datasets, the motion encoders and decoders are trained via a motion reconstruction loss combined with a latent embedding loss at each

quantization layer of the RVQ-VAE structures:

$$\mathcal{L}_{rvq} = \|\mathbf{Z} - \hat{\mathbf{Z}}\|_1 + \beta \sum_{q=1}^{Q} \|\mathbf{z}^q - \text{sg}[\hat{\mathbf{z}}^q]\|_2^2, \qquad (2)$$

where $\text{sg}[\cdot]$ denotes the stop-gradient operation, and $\beta$ a weighting factor for embedding constraint. This framework is optimized with straight-though gradient estimator [44], and our codebooks are updated via exponential moving average and codebook reset following T2M-GPT [54]. After training, the motion encoder and decoders are frozen in the rest of the training process.

### 4.2 Prompt-Motion Alignment Pre-Training

The target of prompt-motion alignment pre-training is to obtain the prompt-motion alignment embedding space, which consists of the contrastive text encoder in Figure 2 and an additional contrastive motion encoder. These two encoders play the essential role to address the issue of missing prompt annotations for speech-to-motion data by employing the *implicit label* strategy. During downstream training, assume that the prompt annotation is needed for some speech-to-motion instance, which is lacking in the original dataset. We can directly input the motion into the contrastive motion encoder and get its corresponding embedding in the prompt-motion aligned space. It is easy to see that this motion embedding is an ideal substitution of the missing prompt embedding if the aligned space is well-trained.

Concretely, motivated by [34], we formulate this pre-training task as a contrastive learning problem. Besides the contrastive text and motion encoders, we employ an additional motion decoder, which is different from the motion encoder in our final inference model. On the premise that the latent space is a probabilistic space,

this setup aims to bring the feature vectors of corresponding text and motion pairs as close as possible. The decoder then decodes these latent feature vectors into motion to calculate the reconstruction loss with real motions. The loss gradients are back-propagated to update the prompt and motion encoders. This technique has been proven to be highly effective in numerous studies [33, 34, 41].

**Loss function design.** We introduce the same set of sub-loss terms to [34], and the total loss can be defined as the weighted sum formulation $\mathcal{L}_{\text{CON}} = \mathcal{L}_{\text{R}} + \lambda_{\text{KL}}\mathcal{L}_{\text{KL}} + \lambda_{\text{E}}\mathcal{L}_{\text{E}} + \lambda_{\text{NCE}}\mathcal{L}_{\text{NCE}}$. For sub-losses, the reconstruction loss $\mathcal{L}_{\text{R}}$ measures the motion reconstruction given prompt or motion input (via a smooth L1 loss). The Kullback-Leibler (KL) divergence loss $\mathcal{L}_{\text{KL}}$ is to regularize the distances between motion and prompt embedding distributions as well as making them closer to the standard normal distribution. The cross-modal embedding similarity loss $\mathcal{L}_{\text{E}}$ enforces both prompt $z^T$ and motion $z^M$ latent codes to be similar to each other (with a smooth L1 loss). A contrastive loss term $\mathcal{L}_{\text{NCE}}$ additionally uses negatives prompt-motion pairs to ensure a better structure of the latent space. More detailed introductions of these loss terms are included in the appendix.

After training, the contrastive text and motion encoders are frozen and utilized in the downstream generation model training.

## 4.3 Generation Model Training

After the pre-training stage, we obtain two sets of outcomes: the motion encoders and decoders, as well as the contrastive text and motion encoders. Based on the motion encoders obtained from motion representation pre-training, we can map all motions in the global motion distribution to the same compact latent space. Utilizing the contrastive text and motion encoders from the prompt-motion alignment pre-training, for motions without prompt annotations in the speech-to-motion dataset, we can provide them with an implicit label using the contrastive motion encoder. What is essential here is that 1) the motion distribution mismatch problem is addressed for motion representations; 2) all co-speech training data have their corresponding (implicit) prompt annotations.

The training of the generation model mostly follows the standard training process of denoising diffusion models [11, 37]. We train the denoising network $E_\theta$ by drawing random tuples $(Z_0, n, A, P)$ from the training dataset, corrupting $Z_0$ into $Z_n$ by adding random Gaussian noises $E$ to obtain $Z_n$, applying denoising steps to $Z_n$ using $E_\theta$, and optimizing the loss

$$\mathcal{L}_{net} = L1_{smooth}[Z_0 - E_\theta(Z_n, n, A, P)]. \tag{3}$$

Specifically, the latent motion representation $Z_0$ is encoded by the motion encoder with the RVQ-VAE structure, and the prompt embedding is obtained from the implicit labels generated by the contrastive motion encoder. Since the speech audio and speech text transcript always occur simultaneously during speech, we uniformly denote them as $A$ here. $A$ is processed through a temporal convolutional network for feature extraction and to align with the latent motion sequences in the time series.

We utilize the classifier-free guidance [19] to train our model. To strengthen the understanding of the two conditional signals, speech $A$ and prompt $P$, we make the diffusion model to learn under both conditioned and unconditioned distributions during training by randomly setting conditional variables **A** and **P** = **0** for $\eta_a$ and

$\eta_p$. This makes the diffusion model better understand the impact of various conditional signals on the generation results. More details of the training techniques are introduced in the appendix.

## 5 MODEL INFERENCE

Through the multi-stage training process, the generation model is obtained. However, utilizing this for conditional generation is not a straightforward task. Even though an aligned space of motion, speech, and prompt is obtained, precise control and generation still requires carefully aligning generation conditions to local body parts. To achieve this target, we introduce the separate-then-combine generation strategy for manipulating latent codes of the diffusion model for both input conditions and body parts.

**General diffusion-based generation process.** During inference, the diffusion network leverages the sampling algorithm of DDPM [18] to synthesize motions. It first draws a sequence of random latent codes $Z_N^* \sim \mathcal{N}(0, I)$ then computes a series of denoised sequences $\{Z_n^*\}, n = N - 1, \ldots, 0$ by iteratively removing the estimated noise $E_n^*$ from $Z_n^*$. The entire process is carried out in an autoregressive manner.

Sampling from $p(Z_0|n, A, P)$ is done in an iterative manner, according to [18]. In every time step $n$ we predict the clean sample $\hat{Z}_0 = G(Z_t, n, A, P)$ and noise it back to $Z_{t-1}$. This is repeated from $t = N$ until $Z_0$ is achieved.

**Separate-then-combine for conditions.** Motivated by Motion-Diffuse [55] and PIDM [4], we extend our system to allow separated guidance to apply the effect of the conditional signal audio and prompt. To achieve this, from the dimension of conditions, we separate latent codes into the following formulation:

$$\mathbf{Z}_{\text{cond}} = \mathbf{Z}_{\text{uncond}} + \mathbf{w_a}\mathbf{Z}_{\text{speech}} + \mathbf{w_p}\mathbf{Z}_{\text{prompt}}, \tag{4}$$

where $Z_{\text{uncond}} = Z_\theta(Z_n, n, \mathbf{0}, \mathbf{0})$ is the unconditioned prediction of the model, such that both the speech and prompt conditions are set as the all-zero tensor **0**. The audio-guided prediction and the prompt-guided prediction are respectively represented by $Z_{\text{speech}} = Z_\theta(\mathbf{Z_t}, \mathbf{t}, \mathbf{A}, \mathbf{0}) - \mathbf{Z}_{\text{uncond}}$ and $Z_{\text{prompt}} = Z_\theta(\mathbf{Z_t}, \mathbf{t}, \mathbf{0}, \mathbf{P}) - \mathbf{Z}_{\text{uncond}}$. $w_a$ and $w_p$ are guidance scale corresponding to speech and prompt.

**Separate-then-combine for body parts.** Furthermore, we extend our system to allow fine-grained style control on individual body parts. We utilize the diffusion model to generate codes for each body part based on masking. The full-body motion codes $\mathbf{Z}^O \in \mathbb{R}^{O \times (L \times C)}$ is then computed by stacking the motion codes of each body part. At inference time, we predict full-body signal $\{\mathbf{E}^*_{\text{cond,o}}\}_{\mathbf{o} \in O}$ conditioned on a set of style prompts $\{\mathbf{P_o}\}_{\mathbf{o} \in O}$ for every body part, where each $\mathbf{E}^*_{\text{cond,o}}$ is calculated by Equation(4). These body part signals can be simply fused as $\mathbf{E}^*_{\text{cond}} = \sum_{\mathbf{o} \in O} \mathbf{E}^*_{\text{cond,o}} \cdot \mathbf{M_o}$, where $\{M_o\}_{o \in O}$ are binary masks indicating the partition of bodies in $O$. To achieve better motion quality, we add a smoothness item to the denoising direction as suggested by [55],

$$\mathbf{Z}^*_{\text{cond}} = \sum_{\mathbf{o} \in O} \left(\mathbf{Z}^*_{\text{cond,o}} \cdot \mathbf{M_o}\right) + \mathbf{w_{body}} \nabla_{\mathbf{Z}_n^O} \left(\sum_{\mathbf{i,j} \in O, \mathbf{i} \neq \mathbf{j}} \mathbf{Z}^*_{\text{cond,i}} - \mathbf{Z}^*_{\text{cond,j}}\right), \tag{5}$$

where $\nabla$ denotes the gradient operator. $w_{body}$ is set to 0.01.

Afterwards, the following generation procedure follows the normal diffusion generation process as discussed above. By utilizing

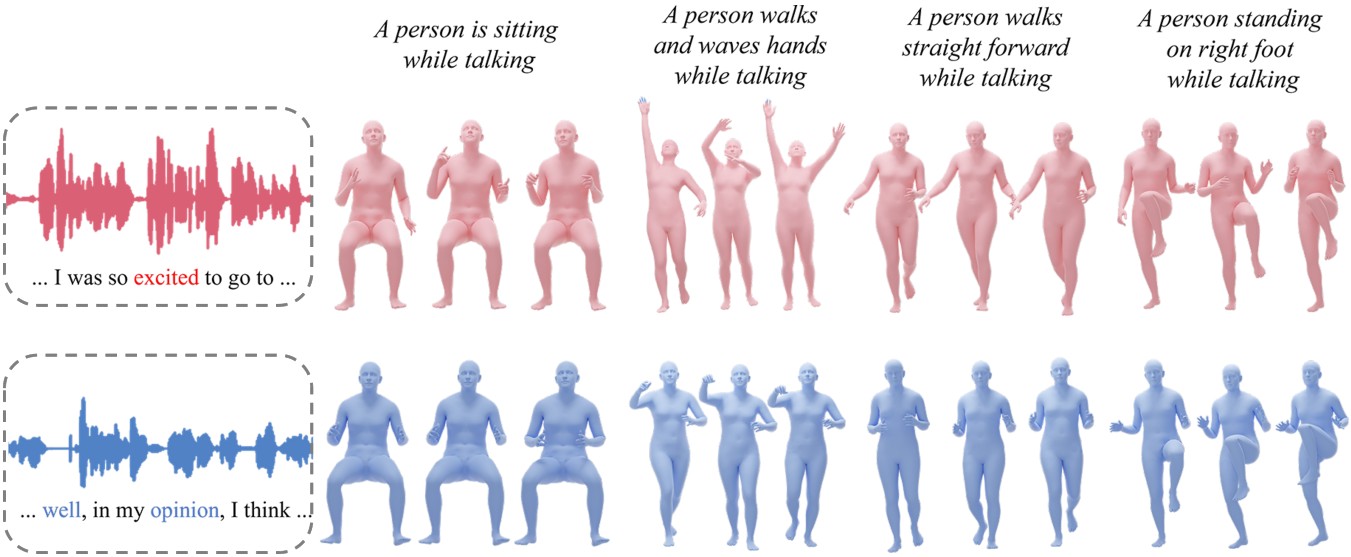

**Figure 4: Qualitative results for synergistic full-body motion generation. More results are included in the appendix as well as demo videos.**

the two separate-then-combine strategies, the control of the motion generation process can be more flexible. The separation steps effectively build more precise mapping between input conditions and body parts. On the other hand, the combine steps ensure to generate synergistic full-body motions, avoiding unnatural motions to appear in generation.

## 6 EXPERIMENTS

In this section, we report experimental results to verify the following questions:

- **Full-body synergistic generation.** Is SynTalker able to generate full-body synergistic motion desirably aligned with both speech and prompt inputs, which is the first co-speech motion generation approach to achieve this functionality?
- **Sinlge-source conditional generation.** Is SynTalker able to achieve comparable or even better performance over state-of-the-art approaches under single-source conditional generation, verifying that besides motions, SynTalker truly learns desirable representations for both speeches and prompts?
- **Ablation study.** Is the multi-stage training strategy and the separate-then-combine inference strategy indeed essential to achieve desirable performance?

Below we explore all three questions in details. More experimental results can also be found in the appendix.

### 6.1 Experimental Setup

**Dataset**. Our model smartly avoids the need for annotated co-speech motion data by leveraging existing speech-to-motion and prompt-to-motion datasets. For the speech-to-motion dataset, we utilize BEATX-Standard [26], which includes 30 hours of co-speech motion, paired with audio and transcripts. For the prompt-to-motion dataset, we employ HumanML3D [16], which annotates motion in

the AMASS dataset [30], consisting of 14,616 annotated motion sequences and 44,970 annotations. All motions are in SMPLX format [32] and consist of 30 frames per second.

**Implementation Details**. We utilize the RVQVAE [53] as our auto-encoder architecture, featuring resblocks in both the encoder and decoder with a downscale factor of 4. Residual quantization employs 6 quantization layers, each with a code dimension of 512 and a codebook size of 512, with a quantization dropout ratio set at 0.2. During contrastive pre-training, we establish a space with a dimension size of 256 and a batch size of 32. We also set the temperature $\tau$ to 0.1, contrastive loss weight to 0.1, and negative-filtering threshold to 0.8. Our diffusion model incorporates 8 transformer layers and is trained with a batch size of 200 and a latent dimension of 512. The number of diffusion steps is 1000. All components can be trained on a single 4090 GPU within three days.

### 6.2 Full-Body Synergistic Generation

In the first experiment, we aim to verify whether our approach effectively supports the generation of synergistic full-body motions conditioned on both speech and flexibly-chosen prompts. As no previous research has addressed this specific task, we focus on evaluating the generation results of our approach using carefully designed input speeches and prompts. It's important to note that we also provide experimental analysis on GestureDiffuseCLIP [2] and FreeTalker [50] in the appendix. These works address related but distinct tasks. GestureDiffuseCLIP also supports prompt-based co-speech motion control. However, this control is limited to motions *inside of the speech-to-motion dataset*. In comparison, our approach supports general out-of-distribution motions. FreeTalker, on the other hand, trained a model capable of switching between speech-to-motion and text-to-motion generation tasks, but it does not produce synergistic results under both speech and prompt conditions.

Table 1: Comparison with the state-of-the-art methods on HumanML3D [16] test set. We compute standard metrics following [16]. For each metric, we repeat the evaluation 20 times and report the average with 95% confidence interval. For MDM and MLD, we report the results using ground-truth motion length.

| Methods | R-Precision ↑ | | | FID ↓ | MM-Dist ↓ | Diversity ↑ |
|---|---|---|---|---|---|---|
| | Top-1 | Top-2 | Top-3 | | | |
| Real motion | $0.490^{\pm.003}$ | $0.682^{\pm.003}$ | $0.783^{\pm.003}$ | $0.001^{\pm.001}$ | $3.378^{\pm.007}$ | $10.471^{\pm.083}$ |
| MDM [42] | $0.363^{\pm.007}$ | $0.553^{\pm.008}$ | $0.662^{\pm.007}$ | $1.390^{\pm.088}$ | $4.599^{\pm.037}$ | $10.704^{\pm.066}$ |
| T2M-GPT [54] | $0.433^{\pm.003}$ | $0.615^{\pm.002}$ | $0.716^{\pm.003}$ | $0.564^{\pm.012}$ | $3.867^{\pm.008}$ | $10.558^{\pm.083}$ |
| MLD [8] | $0.429^{\pm.003}$ | $0.613^{\pm.003}$ | $0.717^{\pm.002}$ | $0.963^{\pm.029}$ | $3.898^{\pm.012}$ | $10.401^{\pm.096}$ |
| MoMask [15] | $0.461^{\pm.002}$ | $0.657^{\pm.003}$ | $0.760^{\pm.002}$ | $0.222^{\pm.007}$ | $3.620^{\pm.011}$ | $10.621^{\pm.096}$ |
| SynTalker (w/o prompt-to-motion alignment) | $0.429^{\pm.003}$ | $0.622^{\pm.004}$ | $0.732^{\pm.004}$ | $0.509^{\pm.013}$ | $4.033^{\pm.013}$ | $10.231^{\pm.096}$ |
| SynTalker (w/o motion representation pre-training) | $0.097^{\pm.002}$ | $0.178^{\pm.002}$ | $0.253^{\pm003}$ | $17.797^{\pm.056}$ | $7.146^{\pm.010}$ | $6.127^{\pm.057}$ |
| SynTalker | $0.375^{\pm.003}$ | $0.564^{\pm.003}$ | $0.681^{\pm.002}$ | $4.385^{\pm.034}$ | $4.499^{\pm.012}$ | $9.374^{\pm.073}$ |

Table 2: Comparison with the state-of-the art methods on BEATX [26] test set. Quantitative evaluation on BEATX. We report FGD $\times10^{-1}$, BC $\times10^{-1}$, and diversity.

| Method | FGD ↓ | BC ↑ | Diversity ↑ |
|---|---|---|---|
| GT | 0.000 | 6.897 | 12.755 |
| recons | 1.729 | 7.122 | 12.599 |
| recons(w/o residual) | 3.913 | 6.758 | 13.145 |
| S2G[13] | 25.129 | 6.902 | 7.783 |
| Trimodal[52] | 19.759 | 6.442 | 8.894 |
| HA2G[28] | 19.364 | 6.601 | 9.671 |
| DisCo[25] | 21.170 | 6.571 | 10.378 |
| CaMN[27] | 8.752 | 6.731 | 9.279 |
| DiffStyleGesture[48] | 10.137 | 6.891 | 11.075 |
| Habibie *et al.*[17] | 14.581 | 6.779 | 8.874 |
| TalkShow[51] | 7.313 | 6.783 | 12.859 |
| EMAGE [26] | 5.423 | 6.794 | 13.057 |
| SynTalker (w/o mo.rep.) | 5.759 | 7.181 | 10.731 |
| SynTalker (w/o align.) | 5.242 | **8.010** | **13.521** |
| SynTalker (w/o both) | **4.687** | 7.363 | 12.425 |
| SynTalker | 6.413 | 7.971 | 12.721 |

As shown in Figure 4, we conduct qualitative experiments to evaluate the synergistic generation results of our model. To better demonstrate that the generation results synergistically integrate both speech and text prompt guidance, we present outcomes under two distinct speech audios: one *excited* and the other *calm*. We evaluate our method using four different text prompts: sitting, waving while walking, standing on the right foot, and walking straight forward. The results show that our model produces talking motions that closely align with the input speech audio while accurately adhering to the text prompt requirements for body gestures. With the excited audio, the motions exhibit more pronounced changes compared to the calm speech. These include increased arm movements, higher arm raises, more pronounced left-right body turns with larger arm movements, and a tendency for hands to reach outward while talking. For additional results, please refer to the appendix.

## 6.3 Single-Source Conditional Generation

In the second experiment, we focus on verifying whether our approach indeed learns a desirable joint embedding space, in special for the input conditions. To achieve this purpose, we introduce two single-source conditional generation benchmarks, i.e. speech-to-motion generation without prompts and prompt-to-motion generation without speeches. By quantitative comparison with state-of-the-art approaches under these two distinguished domains, we are able to verify whether our approach successfully distills information from both speech and prompts, meanwhile avoiding interference among them, which would be revealed by performance degeneration in single-condition generation. Note that for our approach, single-source generation is realized by the similar method utilized in generating $Z_{speech}$ and $Z_{prompt}$ in Equation 4. The implementation details of all contenders are included in the appendix.

**Speech-to-motion.** We compare our approach with state-of-the-art speech-to-motion generation approaches, whose results are cited from [26]. As shown in Table 2, our method significantly outperforms baselines in terms of FGD [52], BC [24], and diversity [23]. This result provides convincing proof that our approach generates significantly desirable speech representation to support strong speech-to-motion generation. To further verify how speech representations are affected by the multi-stage training process, we also conducted ablation studies under this task. The details are discussed in Section 6.5.

**Prompt-to-motion.** In this task, we compare our method with four state-of-the-art text-to-motion generation approaches. The results are reported in Table 1. It can be observed that our approach could achieve comparable performance to the existing baselines, showing its effectiveness in understanding text prompts. Similar to speech-to-motion, we conduct ablation studies for further justification, whose results are reported in the next subsection.

## 6.4 Ablation Study

In this section, we demonstrate qualitative examples of ablation study on model components and assess their contribution to the

synergistic generation capability. For clarity, we demonstrate the results using a single prompt. Please refer to the appendix for additional results. As shown in the Figure 5, given the same speech and the text prompt "a person is talking while sitting", compared to the sitting and talking motion in Figure 5(a), removing our proposed components result in non-ideal generation results. Figure 5(b) corresponds to the removal of *implicit labeling* in train stage, Figure 5(c) corresponds to the removal of *separate-then-combine strategy* in inference stage, and Figure 5(d) corresponds to the removal of *motion representation pre-training* in pre-training stage.

**Implicit labeling.** Figure 5(b) demonstrates the impact of removing implicit labeling during the training stage. Without implicit labeling, the model defaults to merely reacting to the textual prompt, producing only a static sitting motion. This confirms that without our proposed implicit labeling method, the diffusion model does not automatically learn to synthesize and integrate input to produce synergistic motions conditioned on both signals.

**Separate-then-combine strategy.** Figure 5(c) illustrates the effect of omitting the separate-then-combine strategy. Although the model learns to respond to both audio and prompt signals when trained with the implicit labeling method, the textual prompt inherently imposes different requirements on various body parts. The absence of the separate-then-combine strategy eliminates part-level guidance, leading the diffusion model to incorrectly merge multiple features. In this scenario, the model misinterprets the instruction to sit as merely lowering the arms and slightly bending the legs, rather than sitting.

**Motion representation pre-training.** Figure 5(d) shows the results when the joint training stage is omitted from the training process. The character shows an inclination to sit, but such motion is not represented within the limited distribution of the speech-to-motion datasets, rendering accurate generation unfeasible.

**Single-condition experiment ablation.** Though our work focuses on the synergistic co-speech motion generation that follows audio and text motion prompts at the same time, we also evaluate the impact of our proposed methods on the single-source condition generation ability, whose results are shown in both Table 1 and 2.

We first remove all our propose components for synergistic co-speech motion generation, which results in a pure co-speech motion generation model conditioned on audio signal. This base model achieves the state-of-the-art performance in pure co-speech motion generation task, which serves as a solid base for our synergistic generation. Note that our base model only performs co-speech motion generation and it is unable to operate on text-to-motion task, resulting in one less ablation result in Table 1.

We then evaluate the result with prompt-motion alignment while motion representation pre-training is removed. This removal has a positive impact on the text-to-motion generation, as prompt-motion alignment enables the model to accept both the audio and text prompt as input, which drags the model's output distribution to co-speech generation instead of pure prompt-based motion generation.

Finally, we evaluate the single-condition performance when prompt-to-motion alignment is removed while motion representation pre-training is kept. This results in significant performance drop in text-to-motion generation due to the lack of solution space in text-to-motion generation. In co-speech motion generation, this

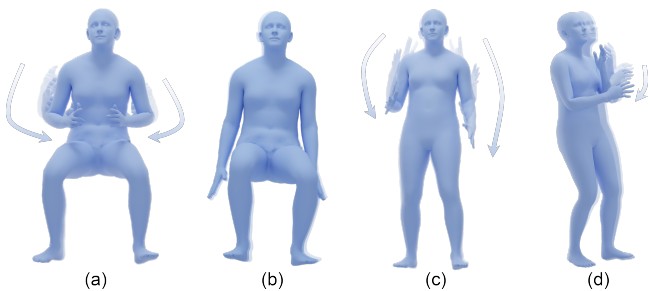

<p>(a)      (b)      (c)      (d)</p>

**Figure 5: Qualitative ablation studies on training and inference procedures. More results are included in the appendix.**

results in a minor performance drop due to the solution space expanding beyond the original co-speech motion distribution, which is desirable in generating synergistic co-speech full body motion.

## 7 LIMITATIONS AND FUTURE WORK

Our method necessitates processing noise and performing parallel inferences at every diffusion step due to the separate-then-combine strategy. On a single RTX 4090, it achieves a generation rate of 10 frames per second without speed-up strategies and 200 frames per second with scheduling methods like DDIM [39]. While this is sufficient for real-time streaming generation applications, our method incurs a higher inference cost than a regular latent diffusion model with the same number of parameters, leading to increased computational demands during deployment.

Although our model implements semantically correct body-level control through the separate-then-combine strategy, it primarily treats conditional inputs as signals of varying strength rather than fully comprehending them. This limitation highlights the inherent challenge in the field of motion generation: the difficulty of generative models in accurately interpreting multi-modal conditions. This difficulty underscores the need for future research to focus on the deeper comprehension of user prompts, which remains a formidable and crucial challenge. Additionally, hand gestures play a significant role in the realism and expressiveness of generated gestures. Currently, the lack of an annotated hand gesture dataset limits the ability to control gestures via textual prompts, which can see significant improvements through future related work.

## 8 CONCLUSION

In this paper, we propose *SynTalker*, targeting at addressing the lack of elaborate control issue of current co-speech motion generation approaches. Our main contributions are: 1) By introducing a multi-stage training process, we effectively utilize off-the-shelf text-to-motion datasets to enable the diffusion model to simultaneously understand both co-speech audio signals and textual requirements. This approach allows for the generation of synergistic full-body co-speech motions; 2) A separate-then-combine strategy during the inference stage, enabling fine-grained control over different local body parts. Extensive experiments demonstrate the effectiveness of our method and show that it can achieve precise control over the generated synergistic full-body motions, surpassing the capabilities of existing methods.

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
