# OpenReview forum: "Enabling Synergistic Full-Body Control in Prompt-Based Co-Speech Motion Generation"
_acmmm.org/ACMMM/2024/Conference — MM2024 Poster_

### Official Review · Reviewer_mAmP · 2024-05-24

**Rating:** 6
**Confidence:** 2

**Summary:**

The paper proposes a prompt-based co-speech full body motion generation pipeline named SynTalker. The paper introduces a multi-stage training process to utilize off-the-shelf text-to-motion datasets and a novel separate-then-combine strategy for inference. Experimental results demonstrate that SynTalker achieves precise and flexible control over full-body motion, surpassing previous methods. Additionally, the video in the supplementary material shows that the generated motions are coherent and well-aligned with the rhythm of the speech.

**Strengths:**

1.  SynTalker is the first approach to enable synergistic full-body control with general text prompts for co-speech motion generation. It addresses the challenge of lacking fully annotated datasets of speech, text, and motion.
2.  The proposed multi-stage training approach effectively handles the motion distributional mismatch and prompts annotation lacking challenges.
3.  The proposed inference strategy allows for precise control over different body parts while achieving synergy among them.
4.  Extensive experiments demonstrate the effectiveness of SynTalker in generating synergistic full-body motions and achieving desirable performance in both speech-to-motion and prompt-to-motion generation tasks.

**Limitations:**

The inference time is not suitable for realtime motion generation.

**Suitability:**

3

---

### Official Review · Reviewer_bLbF · 2024-05-25

**Rating:** 2
**Confidence:** 3

**Summary:**

The paper proposes SynTalker, an approach that uses both speech and text prompts for generating co-speech motion with integrated full-body control. Distinguishing itself from previous works, this study focuses on full-body control by utilizing a multi-stage training process and a unique separate-then-combine strategy during inference. However, since no additional pair-wise text-to-body motion data is provided, the authors rely on existing datasets, which leads to less satisfactory results. Furthermore, the conditional generative model proposed in this paper lacks novelty, as it primarily involves a straightforward integration of existing motion generation techniques.

**Strengths:**

1. Diverging from previous co-speech motion generation approaches focus mainly on upper body gestures, the proposed method focuses on full-body control by utilizing a multi-stage training process and a unique separate-then-combine strategy during inference.
2. Extensive experiments demonstrate the superiority of SynTalker over existing methods, showing its capability to generate nuanced and contextually appropriate motions based on both speech and text inputs
3. The figure in manuscript is helpful for paper reading and understanding.

**Limitations:**

1. The paper title uses the phrase "Synergistic Full-Body Control," but it’s unclear how synergy is achieved between each modality or body part. From my perspective, it appears to be a simple multi-control generative model.
2. No additional pair-wise text-to-body motion data is provided, and the authors rely on existing datasets, which results in less satisfactory outcomes.
3. From the videos in the supplementary material, the visualization results are poor. It might be beneficial to try using a state-of-the-art audio motion generation model combined with a lower body generation model conditioned on the upper body. Such a comparison would be necessary.
4. I noticed that while this model also generates hand poses, the videos do not display any noticeable diversity in the hand gestures.

**Suitability:**

2

---

### Official Review · Reviewer_cT77 · 2024-06-02

**Rating:** 4
**Confidence:** 3

**Summary:**

This paper proposes the first approach to enable synergistic full-body control with general text prompts for co-speech motion generation,
under the situation of lacking fully annotated datasets of speech, text, and motion. It desgin the a novel multi-stage training approach to address the motion distributional mismatch and prompt annotation lacking challenges. In inference phase, it also proposes a novel separatethen-combine approach to achieve both precise control and synergistic motion generation.

**Strengths:**

1. Significant paper quality and sufficient supplementary material.
2. Good writing and easy to understand.
3. Simple motivation for utilizing off-the-shelf text-to-motion datasets to augment co-speech training for obtaining prompt-based co-speech motion generation.

**Limitations:**

My rating are borderline and will be adjusted after authors response.

Although this work utilize the prompt for controling whole body more precisely for the first time, it appears to be intended for use, lacking deep insights.

1. Most problems occurs in introduction. As you mention "One of the fundamental challenges here is that the speech signal is too weak to uniquely determine full-body motions.", What's the meaning of "weak speech signal"? Although the following example can reflect the main motivation about the whole-body controls, it seems cherry-picking cases. I wonder whether give more instruction in speech  like "talking while walking" instead of only "talking" can generate the motion that is more in line with the scene. Otherwise, it easily causes the different motion like you have said in row 112-113.  I want to see some examples.  After that, this paper raises the main problem that current co-speech motion generation approaches usually focus on upper-body gestures but ignore low-body movements due to the flawback of exist datasets like BEATX. Can you point out how serious the problem is by using some statistics skills? It also help me to understand the impact of the mismatch problems.

2. The authors emphasize the proposed method can supports general out-of-distribution and synergistic motion. Authors should give more failure cases about the previous model like FreeTalker by utlizing "out-of-distribution" motions, which is missing in the supplementary videos not only just two cases in appendix. Total qualitative results only support the performance of the model, but not support the comparison. If possible, I recommend authors should give the project websites for detailed illustration.

3. More analysis of the experiments in Table 1/2. For example, for Top-1 in the Table 1, why the SynTalker is lower than others methods except for the MDM ? Similar results also has shown in other terms like FID. It seems the proposed model performance is worse compared to other SOTAs. The author should focus on the whole-body control of text prompt especially in "out-of-distribution" motions.

**Suitability:**

3

---

### Meta-Review · Area_Chair_XprE · 2024-07-02

**Recommendation:** Accept (Poster)
**Confidence:** 5

**Metareview:**

Overall, this paper presents a significant contribution with solid evaluations. Highlights include:

    + Unlike existing literature, it focuses on full-body gesture generation from speech-prompted information, even with limited data.
    + To achieve this, the authors propose a multi-stage training process and a unique separate-then-combine strategy during inference.
    + Extensive experiments demonstrate the effectiveness of the proposed approach.
    + The paper is an excellent fit for presentation at ACM MM, combining human motion, speech, and text.

However, as noted by the authors, there are issues in the evaluation part, which lead me to recommend a poster presentation. These issues can be summarized as follows:

    - More visual results are needed to fully understand the performance of the approach. Specifically, the authors should consider conducting a user study to present qualitative results and better demonstrate the effectiveness of the approach.
    - More comparative results are necessary. The authors should consider implementing baseline approaches to better highlight the effectiveness of their method. For instance, there is a line of work focusing on simultaneous action generation, e.g., [1]. Although such methods have not been applied to nonverbal gesture generation, to the best of my knowledge, the authors should discuss these works to provide stronger comparisons.

[1] SINC: Spatial Composition of 3D Human Motions for Simultaneous Action Generation